# Artificial Intelligence Algorithms Enable Automated Characterization of the Positive and Negative Dielectrophoretic Ranges of Applied Frequency

**DOI:** 10.3390/mi13030399

**Published:** 2022-02-28

**Authors:** Matthew Michaels, Shih-Yuan Yu, Tuo Zhou, Fangzhou Du, Mohammad Abdullah Al Faruque, Lawrence Kulinsky

**Affiliations:** 1Department of Mechanical and Aerospace Engineering, University of California Irvine, 5200 Engineering Hall, Irvine, CA 92627-2700, USA; mmichae3@uci.edu (M.M.); tuoz3@uci.edu (T.Z.); 2Department of Materials and Manufacturing Technology, University of California Irvine, 5200 Engineering Hall, Irvine, CA 92627-2700, USA; 3Department of Electrical Engineering and Computer Science, University of California Irvine, 2200 Engineering Hall, Irvine, CA 92627-2700, USA; shihyuay@uci.edu (S.-Y.Y.); fangzhod@uci.edu (F.D.)

**Keywords:** guided micro-/nano-assembly, artificial intelligence, dielectrophoresis, electrokinetic assembly

## Abstract

The present work describes the phenomenological approach to automatically determine the frequency range for positive and negative dielectrophoresis (DEP)—an electrokinetic force that can be used for massively parallel micro- and nano-assembly. An experimental setup consists of the microfabricated chip with gold microelectrode array connected to a function generator capable of digitally controlling an AC signal of 1 V (peak-to-peak) and of various frequencies in the range between 10 kHz and 1 MHz. The suspension of latex microbeads (3-μm diameter) is either attracted or repelled from the microelectrodes under the influence of DEP force as a function of the applied frequency. The video of the bead movement is captured via a digital camera attached to the microscope. The OpenCV software package is used to digitally analyze the images and identify the beads. Positions of the identified beads are compared for successive frames via Artificial Intelligence (AI) algorithm that determines the cloud behavior of the microbeads and algorithmically determines if the beads experience attraction or repulsion from the electrodes. Based on the determined behavior of the beads, algorithm will either increase or decrease the applied frequency and implement the digital command of the function generator that is controlled by the computer. Thus, the operation of the study platform is fully automated. The AI-guided platform has determined that positive DEP (pDEP) is active below 500 kHz frequency, negative DEP (nDEP) is evidenced above 1 MHz frequency and the crossover frequency is between 500 kHz and 1 MHz. These results are in line with previously published experimentally determined frequency-dependent DEP behavior of the latex microbeads. The phenomenological approach assisted by live AI-guided feedback loop described in the present study will assist the active manipulation of the system towards the desired phenomenological outcome such as, for example, collection of the particles at the electrodes, even if, due to the complexity and plurality of the interactive forces, model-based predictions are not available.

## 1. Introduction

The art and science of miniaturization forms the foundation for the rapid progress of the last 30 years in materials science, computational technology, in communications, and in biotechnology [1,2,3,4]. In microdevices, from modern transistors to Labs-on-Chip, reduction of the size of the components results in faster operation, cheaper cost per device, reduction in energy budget to manufacture and to operate a device, and higher sensitivity and selectivity for microsensors [5]. Often, the manufacturing bottleneck is the speed and accuracy of assembly of the device from micro- and nano-sized components [6]. Micro-assembly is the positioning, orienting, and assembling of micro-scale components into complex microsystems with the goal of achieving hybrid micro-scale devices of high complexity while maintaining high yield and low cost [7]. Micro-assembly techniques can be characterized into two major categories, serial assembly and parallel assembly.

Serial assembly methods, often referred to as pick-and-place assembly, involve selecting individual parts and moving and placing them in a desired location using various versions of microgrippers, micro-scale tweezer-like instruments capable of capturing, moving, and releasing micro parts [8,9,10]. In recent years, serial assembly technology has continued to develop, including non-contact methods of micro-part manipulation such as through the use of capillary forces [11] or through fluidic forces emanating from vibrating piezo actuators [12].

While serial assembly methods have proven effective at moving and placing individual parts, they suffer from slow assembly speeds and, consequently, low throughput. Research into the use of robotics has improved serial assembly speed; however, these methods remain prohibitively slow for most industrial applications [13].

An alternative strategy for micro-assembly is the use of parallel assembly techniques. Parallel assembly technologies include self-assembly and directed assembly. In self-assembly, micro parts autonomously assemble into microstructures, often triggered by a specific environmental condition [14,15]. A growing area of research within self-assembly is micro-origami, a process in which a 2D micro part folds into a 3D microstructure, including DNA-based nanomanufacturing [16] and magnetic field-induced origami folding of nanomembranes [17].

Directed assembly, which combines the specificity of the placement of the serial assembly and the speed of self-assembly, includes processes which can manipulate and assemble multiple micro- and nano-sized parts in parallel. Directed assembly techniques typically take advantage of forces dominant in the micro and nano domain, such as capillary forces and various charge-based interactions collectively known as electrokinetic forces [18].

Dielectrophoresis (DEP), one of the electrokinetic forces, acts on charged or non-charged dielectric particles placed in a non-uniform electric field [19]. DEP can either cause attraction of the particles towards the electrodes (so-called positive DEP (pDEP)) or repulsion of the particles from the electrodes under the negative DEP (nDEP). Whether the DEP force is positive or negative depends on the sign of the Claussius–Mossotti factor *K* established in Equation (1):(1)$$K=\left(\varepsilon_{p}^{*}-\varepsilon_{m}^{*}\right)/\left(\varepsilon_{p}^{*}+2\varepsilon_{m}^{*}\right)$$
where $\varepsilon_{i}^{*}\;$ stands for complex permittivity of the materials and subscripts *p* and *m* identify particles and suspension media, respectively [19].

Dielectrophoretic force on a spherical dielectric particle, given by Equation (2), depends on the particle radius, *R*, the real part of the Claussius–Mossotti factor *Re*[*K*], and the gradient of the square of the electric field *E*:(2)$$F_{DEP}=2\pi \varepsilon_{m}R^{3}Re\left[K\right]\left(\nabla E^{2}\right)$$

Because the sign of *Re*[*K*] depends on the applied frequency of the AC bias, it is possible to tune the frequency to switch between the negative and positive DEP—i.e., to cause the micro- and nanoparticles that were previously repelled from the electrodes to be attracted to the electrodes (and vice versa).

Therefore, DEP is widely used as a basis for sorting and separating microparticles [20], including biological cells [21]. One such example is in-droplet cell separation technology developed by researchers at Texas A&M University that successfully utilized DEP force to separate a mixture of *Salmonella* and macrophages in order to improve microfluidic cellular assay capabilities [22]. DEP-based technology has also been shown to have application in medical testing and diagnostics such as separating cancerous cells from the healthy cells from a homogeneous mix [23]. Another study details the use of DEP force to separate circulating cancer cells from blood samples. Compared to traditional methods such as anti-body labelling, this research demonstrated various benefits of DEP separation such as the fact that DEP is not dependent on surface markers on the cancer cell surface, making it more generally applicable and resulting in more viable samples of the target cells [24]. In addition to microfluidic applications, the guided electrokinetic micro assembly of microparticles using DEP force to specific locations on an electrode array is a recent promising development that has application in many fields ranging from biotechnology to micro- and nano-electronics [25]. The ability to selectively move large quantities of microparticles to specified locations is a significant improvement over pick-and-place technology and is critical in the development of commercially viable bottom-up micro- and nano-assembly.

In order to successfully implement electrokinetic manipulation of micro- and nanoparticles, we need to model the physical forces present in the microfluidic system and thus predict the specific applied frequency required to cause the positive or negative DEP for a specific population of particles. Two issues make modeling and prediction difficult. The first issue is that there are several parameters of the microparticles (including their exact electrical permittivity and conductivity) that are difficult to determine experimentally and often DEP experiments are performed to indirectly determine the electrical conductivity of micro- and nano-particles based on the experimentally determined crossover frequency (the frequency at which positive DEP switches to negative DEP) [26]. Another issue concerns the magnitude of the various forces acting on the microparticles. Unlike the macro scale where one force is often dominant, such as gravity, there are often multiple competing forces acting on parts on the micro/nano scale, including dielectrophoresis, electroosmosis (EO), particle-to-particle interaction, viscous drag, natural convection, etc. [18]. These competing and inter-related forces make the physical modeling of electrokinetic separation and propulsion of micro- and nano-particles extremely challenging.

The present work describes an artificial intelligence (AI)-based phenomenological approach that automatically determines the ranges of the applied frequency that cause positive and negative DEP for the population of 3-μm diameter polystyrene beads in deionized water. Specific crossover frequency can be calculated from Equation (1) setting the real part of K to be equal to zero. In this system, the electrical conductivity and permittivity of DI water and polystyrene beads would be required for such calculation. From the literature, we know the relative permittivity of DI water to be around 78 and relative permittivity of the latex beads to be around 2.5, while the electrical conductivity of DI water to be around 2 × 10^−4^ S/m [27]. The problem confronts the researchers trying to gauge the conductivity of the polystyrene microbeads, since polymer beads’ conductivity is highly dependent on their functionalization and bead radius. The beads’ electrical conductivity is not measured directly. Typically, the conductivity of the beads is deduced indirectly by observing the crossover DEP frequency and back-calculating the conductivity of the beads from Equation (1) [28]. Therefore, the described phenomenological approach of finding the crossover frequency is critically important because it cannot be predicted beforehand, but can only be observed directly through experimentation.

Our AI-based automated approach involved a custom-developed closed-loop system where the function generator (that applied AC signal to the electrodes) was connected to the computer and the Python program was used to change the applied frequency of the signal. The movement of the microbeads was digitally captured by the camera attached to the microscope and the resulting images were automatically analyzed to determine if the beads are moving towards the electrodes or away from the electrodes. On the basis of that determination, the applied frequency used by the function generator was algorithmically changed to a new frequency value and the analysis was repeated. The outcome of the automated analysis is the frequency ranges where DEP force is positive, negative, or too weak to cause significant particle motion.

The phenomenological approach to directed micro-assembly detailed in this research will find application across many fields including microsystems and electronics, biotechnology, drug delivery, and tissue engineering.

## 2. Materials and Methods

### 2.1. Interdigitated Electrode Fabrication

The gold interdigitated electrodes arrays (IDEAs), used to generate non-uniform electric fields in this study, were fabricated via conventional photolithography and e-beam evaporation. Positive photoresist (Shipley) was spin-coated onto a 4-inch silicon wafer (University Wafer, South Boston, MA, USA) at 3000 revolutions per minute (rpm) for 30 s, after an initial spin-coating at a speed of 500 rpm for 10 s, using a Laurell photoresist spinner (Laurell Technologies, North Wales, PA, USA). Next, the coated wafer was soft-baked on a hot plate (Dataplate, PMC, 732 Series, Dubuque, IA, USA) at 90 °C for 30 min. After soft-bake, the photoresist coated on wafer was exposed to UV light utilizing a MA56 Mask Aligner (Karl Suss, Garching, Germany) for 4 s at an energy intensity of 10 mW/cm^2^, through a photomask (CadArt, Bandon, OR, USA). Exposed photoresist was then removed by the deionized (DI) water rinse.

A Temescal CV-8 E-beam evaporator (Airco Inc., Berkeley, CA, USA) was used to deposit metal layers. Following the deposition of a 300 Å first layer of Cr, a 3000 Å layer of Au was deposited onto the underlying chromium layer. Then, the unexposed photoresist was dissolved in acetone. After this photoresist stripping step, only the metals that covered the wafer, rather than photoresist, were left in the so-called lift-off process. The resulting electrode system, presented in Figure 1, consists of 12 interdigitated fingers separated by the gaps of 70 µm.

### 2.2. Experimental Setup

The IDEA, with wire contacts soldered by indium, was connected to a function generator (Stanford Research System, Sunnyvale, CA, USA), as illustrated in Figure 2. The double-sided adhesive tape (3M, Saint, Paul, MN, USA) was cut to construct a polymer cage to confine liquid with suspension of microbeads. The peak-to-peak voltage and frequencies applied by the function generator are controlled via digital input from the computer via Standard Commands for Programmable Instruments (SCPI) programming language and the Virtual Instrument Software Architecture (VISA) API.

Aqueous solutions of 3 μm diameter carboxyl-modified latex (CML) polystyrene beads (Thermo Fisher Scientific, Invitrogen, Waltham, MA, USA), which was originally at 4 wt % concentration, was diluted to a new concentration of 0.39 wt % in DI water by first placing the original solutions in a centrifuge (Eppendorf, Hamburg, Germany) for 20 min at 2000 rpm, then removing the supernatant with pipette, and eventually remixing the remaining beads with a pre-calculated amount of DI water.

A 10 μL droplet of prepared beads suspension was pipetted onto the region of IDEA surrounded by the polymer cage, then a 3 V peak-to-peak sinusoidal voltage with a sequence of discrete frequencies was applied to study the motion of the beads. A Nikon Eclipse microscope (Nikon, Japan) and SPOT Basic video editing program (SPOT Imaging, Sterling Height, MI, USA) were used to observe and record the movement of the particles.

## 3. Software Architecture and Methodology

### 3.1. Architecture Overview

As Figure 3 illustrates, we have implemented the monitoring of the experimental testbed (a suspension of the microparticles on the IDEA chip) where particles movements are captured by the camera and that information is passed in digital form to the computer, where the particles are recognized, their positions are detected, and computer algorithm is run to compare the mean present location of the particle cloud with the mean location of the particle cloud as captured in the prior frames. Based on the comparison of these mean past and present locations of the particle clouds, the function generator is given a command by the computer to either increase or decrease the applied frequency. Therefore, a real-time control and feedback of the embedded system is implemented that allows for the dynamic control of the movement of the particles. Specifically, our system tracks the overall particle movement by a watching window W which is maintained from the result of Particle Detection and Feature Extraction for each new frame. Our system analyzes W with Particle Movement Determination and Adjustment Determination to determine the amount of adjustment made to the Function Generator. Subsequently, the system applies the predetermined changes (such as step up or step down in the applied frequency) to the experimental testbed. We will describe each of the functional components below.

### 3.2. Particle Detection and Feature Extraction

For each frame $I_{t}\in I$, we utilize the Hough Circle Detector [29] to detect particles from each sliced image $I_{t}$. We leverage the implementation of the Hough Circle Detector function in OpenCV, which can perform the Hough Gradient Method on detecting the circle-shaped objects on grey-scale images. To ensure the performance of particle detection across different experimental setups, we tune the four parameters (param_1, param_2, min_radius, max_radius) provided by OpenCV’s implementation. Specifically, the parameter param_1 is for adjusting the internal Canny detector threshold while the parameter param_2 is for modifying the center detection threshold. We denote the result of detecting particles as $B_{t}$, which is a container of a collection of detected particles $\left\{P_{1},\;P_{2},\;\ldots P_{N}\right\}$, where N is the total number of detected particles. In this paper, we denote the particle detecting process using function F. Thus, the relation between a new frame $I_{t}$ and a container $B_{t}$ can be written as: $F\left(I_{t}\right)=B_{t}=\left\{P_{1},\;P_{2},\;\ldots P_{N}\right\}$, where each particle is represented with the center 2-D coordinates (i.e., $P_{n}=\left\{x_{n},\;y_{n}\right\}$).

We then transform each container $B_{t}$ into a corresponding feature $X_{t}.\;$ In our paper, two assumptions are made. First, we assume that the movement of particles (moving toward/away from the electrodes) can be captured and represented by the values derived from formulaic methods. Second, since the direction of DEP force is perpendicular to the electrodes installed in our testbed, we only consider a particle’s movement along the *x*-axis in our system. In our paper, we abstract this feature extracting process using a function G. In our system, we implement G by calculating the average absolute distance to the reference line where the reference line refers to a vertical line that stays in between and in the middle of two electrodes. We denote the *x*-coordinate of the reference line as r and use it along with the *x*-coordinate of each particle in a container $P_{n}=\left\{x_{n},\;y_{n}\right\}\;\in \;B_{t}$ is used to calculate the feature $X_{t}$. In short, the relationship between a new frame $I_{t}$ to the extracted feature $X_{t}$ is illustrated as:(3)$$X_{t}=G\left(F\left(I_{t}\right)\right)=G\left(B_{t}\right)=\sum_{n=1}^{N}\frac{\left|x_{n}-\;r\right|}{N}\;$$

### 3.3. Particle Movement Determination

In order to determine the overall movement of particles at each timestamp t, our system performs a linear trend analysis by considering a subset of features between the current frame and few frames prior. We define this set as a watching window $W=\left\{X_{t-k},\;X_{t-k+1},\;\ldots X_{t}\right\}$, where *k* denotes the length of *W* and is a tunable system parameter in our system. Our particle movement determination process begins with post-processing the features with Missing-Value Sampling and Data Smoothing. As our particle detector might end up not detecting any particle for $I_{t}$, we need an alternative value of $X_{t}$ for minimizing the negative influence of missing values in our linear trend analysis. To be more specific, we sample the missing value by averaging the features *u* frames before $X_{t}$ ($\left\{X_{t-u-1},\;X_{t-u},\;\ldots X_{t-1}\right\}$), where u≤k. The determination of u depends on the velocity and frame rate of the testing setup.

Next, we perform the Data Smoothing on W to decrease the influence of noise or random errors and to acquire a cleaner trend during our linear trend analysis later. Specifically, we perform linear convolution on $W=\left\{X_{t-k},\;X_{t-k+1},\;\ldots X_{t}\right\}$ with an unweighted filter. The 1′s array performs as an unweighted filter and *u* is a tunable parameter for manipulating the level of smoothing.

Finally, our Particle Movement Determination adopts a Linear Trend Model (LTM) in determining the overall movement of particles. In our system, we perform LTM over the features $\left\{X_{t-k},\;X_{t-k+1},\;\ldots X_{t}\right\}$ in the watching window W to derive the trend of the data. Specifically, we first convert the features in W into a set of data points where each data point corresponds a timestamp x to a feature $X_{t-k+x}$ (e.g., (x, y) = (0, $X_{t-k}$), (1, $X_{t-k+1}$), …, (k, $X_{t}$)). Then, LTM applies the least-squares regression to generate a unique trend line represented by equation y=bx+c that minimizes the vertical distance over the data points, and the coefficient, b, is calculated according to Equation (4):(4)$$b=\frac{\sum \left(x-\bar{x}\right)*\left(y-\bar{y}\right)}{\sum {\left(x-\bar{x}\right)}^{2}}$$

The coefficient b represents the velocity (e.g., the slope of the regression line) of overall particle movement. The system uses b and a decision threshold δ to classify the macroscopic motion of particles with a categorical label $Y_{t}$. If |b|≤ δ, the system classifies the motion as NO_DEP because b is too little to be considered as a DEP polarity. In the case where |b|>δ, the system classifies the DEP’s polarity as either Positive-DEP or Negative-DEP, according to the practical experimental setup (i.e., can be flipped).

### 3.4. Feedback Control Design

The system we propose is a real-time closed-loop controlling system, also known as a feedback control system. The proposed feedback control design is illustrated in Figure 3. In our system, the process variable is the motion of particles. Therefore, with a predefined sampling rate *m*, our system repeatedly collects the new frame from the sensory camera and performs the Particle Detection and Feature Extraction for each frame $I_{t}$ to acquire a feature $X_{t}$. The new feature $X_{t}$ is then inserted to the end of the watching window W, which is realized as FIFO (First-In First-Out) Queue with k as the size in our system. Next, if not in SETTLE state, our system performs the Particle Movement Determination over the watching window $W=\left\{X_{t-k},\;X_{t-k+1},\;\ldots X_{t}\right\}$ and acquires $Y_{t}$, indicating the current state of the testbed. Then, according to $Y_{t}$ and b, our system calculates the amount of frequency or voltage to be adjusted to the testbed.

Ultimately, these adjustments are encapsulated as a command packet and sent to the function generator. Once the function generator is adjusted, our system will enter the SETTLE state for a period. SETTLE is the time required for the system to detect the change in movement of particles due to the function generator change, and it is calculated based on two values: the particle response time and the system response time. The particle response time refers to the time needed for particles to show the effect of a function generator change, and the value depends on the environmental setup. System response time refers to the time needed for the proposed computer-vision-based system to detect the movement of particles, which depends on the length of the watching window, *k*, as we discussed in the section above. In general, a larger *k* will require longer system response time. In SETTLE state, our system just keeps acquiring inputs from the camera and performing Particle Detection and Feature Extraction. In non-SETTLE state, our system will additionally make decisions and adjustments according to the results from Particle Movement Determination. Pseudo code for the control loop is included in Algorithm 1 below.
**Algorithm 1** Pseudo code governing the control loop.
***Data****: particles, watching_window, detected_movement, not_move_count system initialization;***1*****while****not_move_count < threshold do***2**
  *Detect particles in the current frame;*
**3**
  *Append detection results to the watching_window;*
**4**
  *Analyze the direction of particle movement and save result to detected_movement;*
**5**
  ***if***
*|detected_movement| < threshold then*
**6**

   *STATE <= NOT_MOVE;*
**7**

   *not_move_count += 1;*
**8**
  ***else if***
*detected_movement > 0 then*
**9**

   *STATE <= POSITIVE_DEP;*
**10**

   *function_generator.decrease();*
**11**
  ***else***
**12**

   *STATE <= NEGATIVE_DEP;*
**13**

   *function_generator.increase();*
**14**
  ***end***
**15** ***end***


## 4. Results and Discussion

### 4.1. Particle Detection

The cyber-physical system was set up and run as detailed in Section 2 and Section 3 above. As the test proceeded, the program converted the live video stream (transferred from the optical microscope via the digital camera) into still-frame images. The OpenCV package, utilizing the Hough Circle Detector function, identified beads within each frame. Figure 4 below shows examples of bead detection.

Based on the analysis of the individual still-frame images, it is estimated that the OpenCV package was successful at accurately identifying an average of 20–30% of the beads in each frame, typically with higher bead detection, as shown in Figure 4. This detection rate provides an adequate sampling of bead position for each frame to estimate overall movement of the beads. Experimentally, bead detection rate and accuracy were found to be dependent on several different factors, including the amount of illumination and focal settings of the optical scope as well as the density and size of the beads. In general, larger beads at lower concentration led to greater bead detection rate and accuracy. This is thought to be due to the fact that the Hough Circle Detector function utilizes pixel color gradient to detect the beads and larger beads provide a larger circumference for the program to detect such gradient. Additionally, lower bead concentration led to lower likelihood of agglomeration formation, which hindered the ability for OpenCV to recognize their shape as circular. Figure 4 presents examples of the beads in close proximity that were not recognized as individual beads. Additionally, given that OpenCV uses color gradient to detect the edges of the bead, optical settings which provided high contrast between the beads and the silicon chip background yielded better particle detection and accuracy. This is illustrated in Figure 5 below.

### 4.2. Bulk Cloud Behavior

The cyber-physical system was set up as detailed in the previous sections and two trials were performed with 3 µm diameter beads. The video segments of the bead movement under the influence of the applied voltage signal with a given frequency is provided as Appendix A accessible online. The program calculated the average absolute distance of all detected beads from the center of the electrode gap for each frame. The results of each trial, overlaid with the changes to the frequency of the applied electric field, are shown in Figure 6 and Figure 7 below.

In this analysis, increasing absolute value from the center of the testbed indicates that the particles are moving toward the edges of the electrodes, a result of attractive forces such as positive DEP and EO. Conversely, decreasing absolute value from the center indicates that the beads are moving away from the surface of the electrodes in response to an induced repulsive force under negative DEP. The graphical results from both trials shown above clearly depict regions of attractive and repulsive force. For example, in both trials, frequency increase from 10 kHz to 1 MHz resulted in a decrease in the beads’ absolute distance, indicating a change from attractive to repulsive force. The bulk response of the beads indicates attraction at lower relative frequency and repulsion at higher frequencies, which aligns with the established literature on DEP force [30]. As the applied frequency approached the crossover frequency, the magnitude of bead movement decreased.

The results above also indicate a delayed bead response to changes in the applied frequency. This is thought to be the result of residual bead momentum that must be overcome prior to a reversal in bead movement. This interesting result is important in understanding the fundamental physics of this DEP-based particle manipulation system. Such time-delayed response would need to be accounted for when designing such a system for a particular application.

### 4.3. Individual Particle Behavior

Still-frame image results from Trial 2 were analyzed to assess the response of individual beads to changes in the frequency of the applied electric field. Figure 8, Figure 9, Figure 10 and Figure 11 below show subsequent still-frame images from select regions of interest within the Trial 2 results. Specific beads are indicated within each frame for clarity.

As the images indicate, individual beads responded to changing input frequency. In Figure 9, the beads move toward the edges of the electrode at an applied frequency of 10 kHz. Additionally, when the frequency was increased to 1 MHz, the same beads can be seen moving away from the electrode, as shown in Figure 11.

### 4.4. Correlation of Bead Cloud Behavior and Individual Bead Position

#### 4.4.1. Bulk Motion

The individual bead analysis correlates closely with the assessment of the cloud bead behavior made by the cyber- physical system. For instance, in Trial 2, the steady decrease in average absolute distance which occurred after the frequency was set to 1 MHz as shown in Figure 7 correlates to individual bead movement toward the center of the electrode gap in Figure 11.

This result supports the assertion that the proposed algorithmic artificial intelligence program can effectively estimate the response of beads to varying input frequency. The proposed system successfully identified frequency regimes in which the beads were attracted to the electrode surface due to positive DEP and EO, as well as regimes in which the beads were repelled from the electrodes due to negative DEP. Results from the system testing revealed a lag in particle movement in response to changes in frequency. This is seen in Trial 1, when the frequency was changed to 500 kHz at frame 700 as compared to the more immediate particle movement at frame 300 when frequency was changed to 1 MHz. This lag is thought to be the result of particle momentum in relation to the strength of the DEP force generated at a given frequency. Note that 1 MHz produces strong negative DEP, which more quickly overcomes the inertia of the moving particle and results in a rapid change in direction. Alternatively, a frequency of 500 kHz produced a weaker negative DEP force. As a result, the plot shows a gradual slowing of the particles before final reversal of direction roughly 100 frames after the change to frequency was made.

#### 4.4.2. Regions of DEP Influence

As the applied frequency approached the theoretical crossover frequency, the DEP forces exerted on the beads becomes weaker, resulting in slower bead movement. Additionally, DEP force is strongest at the surface of the electrode and becomes weaker as distance from the surfaces increases. As a result, as the applied frequency approaches the crossover frequency, the response of beads further from the electrodes was diminished. This can be seen in the relative magnitude of particle motion in the individual still-frame analysis (Figure 8, Figure 9, Figure 10 and Figure 11) as well as the leveling-off of average absolute distance in Figure 7. Analysis of the bulk particle movement as a function of the applied frequency indicates that when the frequency is increased to 1 MHz, the distance to the center decreases sharply, pointing to negative DEP (nDEP) and indicating that the crossover frequency is below 1 MHz. Meanwhile, at 500 kHz, there is still an attractive influence since the distance from the center is increasing (pDEP). Therefore, the crossover frequency is between 500 kHz and 1 MHz. While we cannot calculate the exact crossover frequency, because the surface conductivity of the CML-modified polystyrene beads is unknown, we can conclude that our range of crossover frequency is consistent with the data for crossover frequency of polystyrene microbeads found experimentally by other researchers [26,27].

The cyber-physical system was successful at identifying a frequency range in which little to no bead movement was detected, indicating that the DEP crossover frequency exists somewhere within the range. Additionally, the system successfully identified frequency regimes in which positive and negative DEP force was strong and resulted in associated bead movement. Bead-to-bead interaction and the formation of pearl chains, as described in the following section, form a primary limiting factor in the system’s ability to further refine these regions of DEP influence.

### 4.5. Bead-to-Bead Interaction

Previous research has established that, in addition to DEP and EO force, particles within an applied non-uniform electric field experience bead-to-bead attraction and form what is known as “pearl chains,” which align along the electric field lines [31]. These pearl chains can be seen within the individual still-frame images, such as Figure 12 below.

The formation of pearl chains as a result of bead-to-bead interaction impacts bead response to DEP force. As the pearl chains grow, the motion of such particle chains is inhibited due to inertia and the increased drag force.

## 5. Conclusions

The cyber-physical system presented in this research successfully utilized algorithmic AI and a phenomenological approach to characterize particle response to changes in the frequency of an applied non-uniform electric field. As detailed above, this approach was able to define regions of attraction and repulsion due to DEP and EO force as well as regions of weak DEP force which resulted in no bead movement. The AI-guided platform has determined that positive DEP (pDEP) is active below 500 kHz frequency, negative DEP (nDEP) is evidenced above 1 MHz frequency and the crossover frequency is between 500 kHz and 1 MHz. These results are in line with previously published experimentally determined frequency-dependent DEP behavior of the latex microbeads. The research presented in this study serves as a first proof of concept that the use of AI and closed-loop cyber-physical systems, along with a phenomenological approach, can be used to study the complex forces exerted on bodies within the micro and nano domains. Such technology can be used to enhance current microfabrication techniques, including bottom-up micro- and nano-manufacturing, and may find applications in various fields including drug delivery, micro-sensor fabrication, and bioassays. The phenomenological approach assisted by the live AI-guided feedback loop described in the present study will assist the active manipulation of the system towards the desired phenomenological outcome such as, for example, collection of the particles at the electrodes, even if due to the complexity and plurality of the interactive forces, model-based predictions are not available. Advances in AI recognition of microbeads and various bead formations will further enhance the described phenomenological AI-guided approach to detection and control of the movement of microparticulate.

## Figures and Tables

**Figure 1 micromachines-13-00399-f001:**
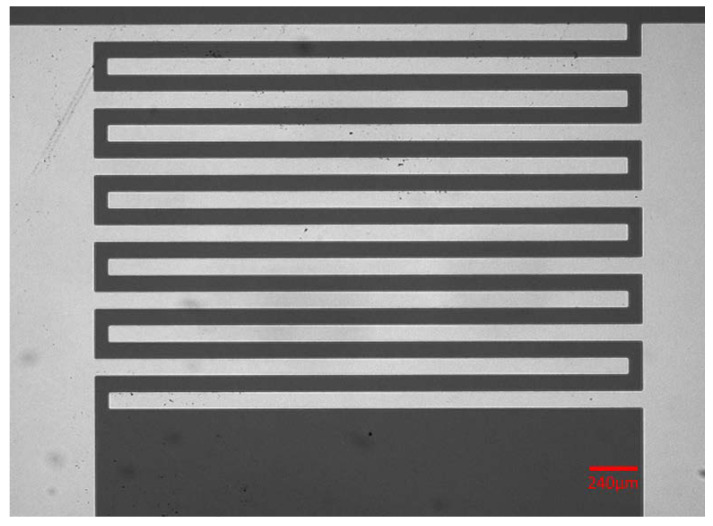
The schematic of the IDEA. The gold electrode fingers (light against the dark background of the substrate) have the spacing between the adjacent fingers of 70 µm.

**Figure 2 micromachines-13-00399-f002:**
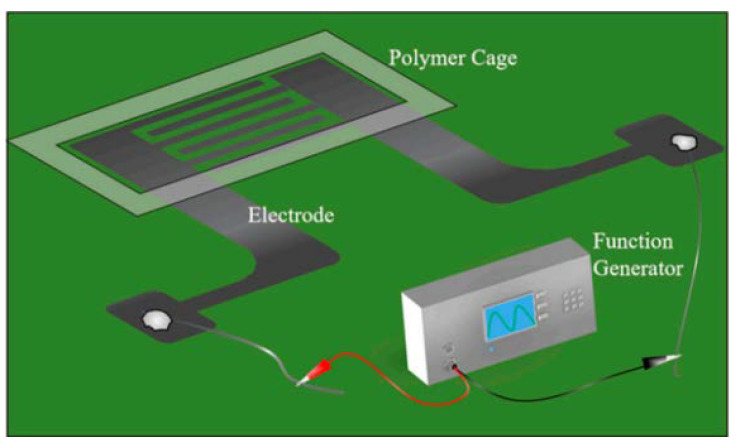
Sketch of the experimental setup including IDEAs connected to a function generator.

**Figure 3 micromachines-13-00399-f003:**
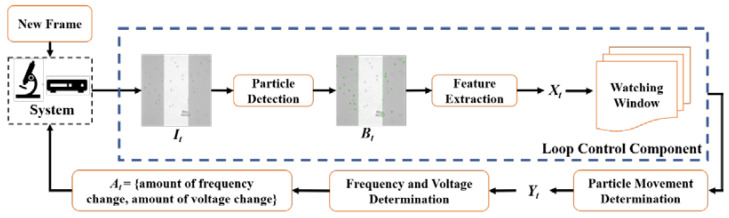
Architecture of the Feedback Control System Design.

**Figure 4 micromachines-13-00399-f004:**
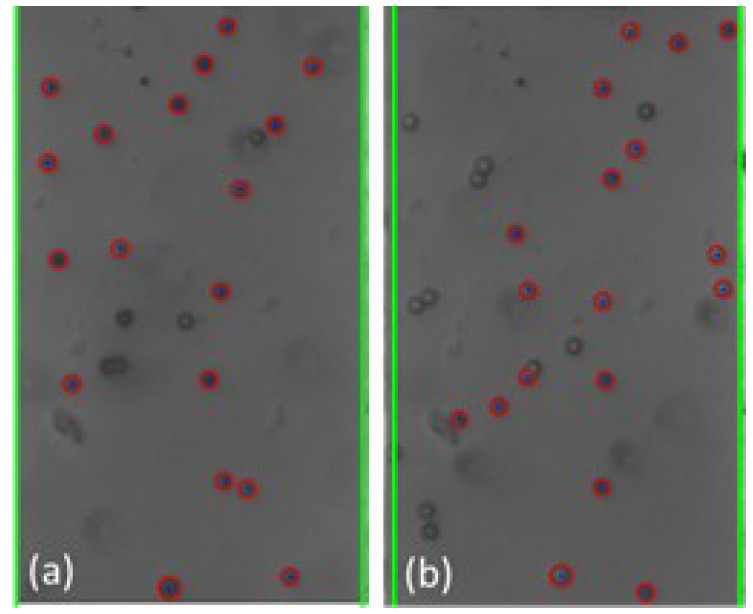
Examples (**a**,**b**) of bead detection using the Hough Circle Detector function of the OpenCV package. The recognized beads are circled in red. The green lines identify the frame window nearly coincident with the edges of the electrodes.

**Figure 5 micromachines-13-00399-f005:**
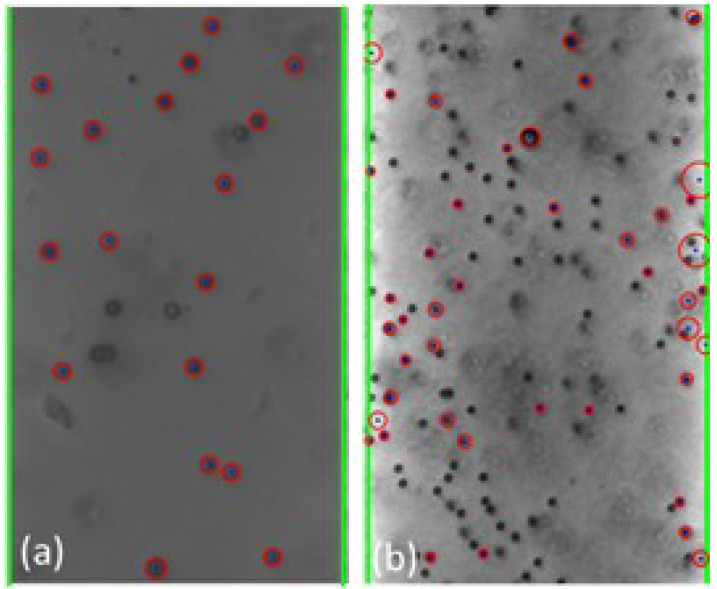
Different bead detection efficacy between samples (**a**,**b**) depended on bead size and illumination conditions.

**Figure 6 micromachines-13-00399-f006:**
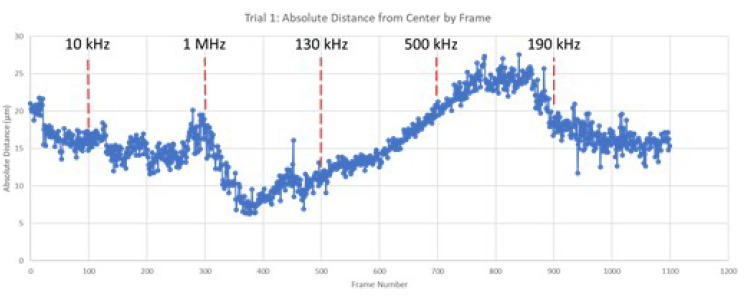
Average absolute distance from center by frame (Trial 1).

**Figure 7 micromachines-13-00399-f007:**
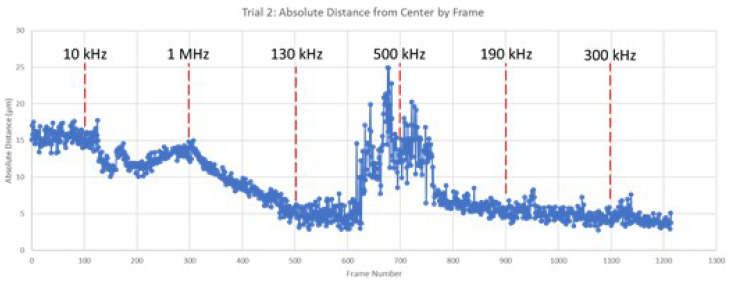
Average absolute distance from center by frame (Trial 2).

**Figure 8 micromachines-13-00399-f008:**
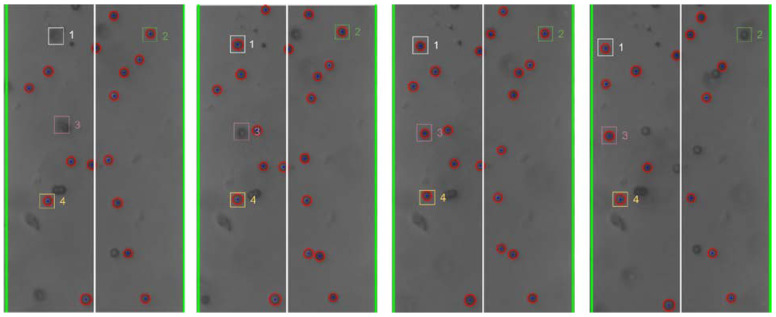
Individual bead movement during Trial 2 from frames 195 to 298.

**Figure 9 micromachines-13-00399-f009:**
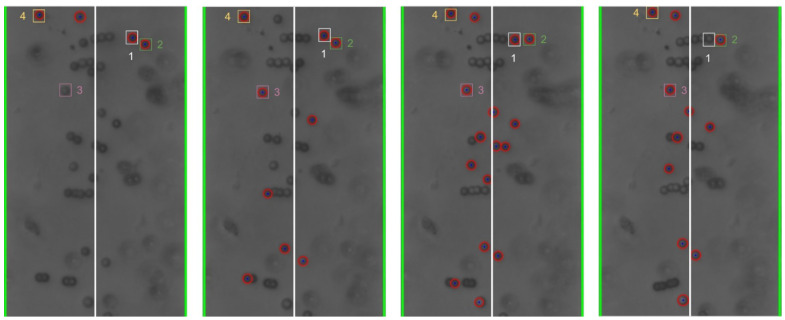
Individual bead movement during Trial 2 from frames 300 to 340.

**Figure 10 micromachines-13-00399-f010:**
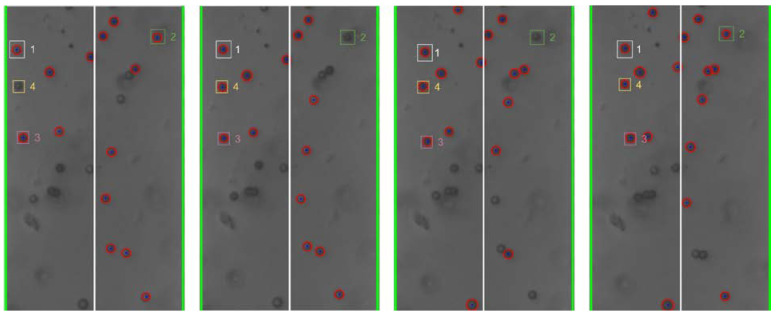
Individual bead movement during Trial 2 from frames 570 to 685.

**Figure 11 micromachines-13-00399-f011:**
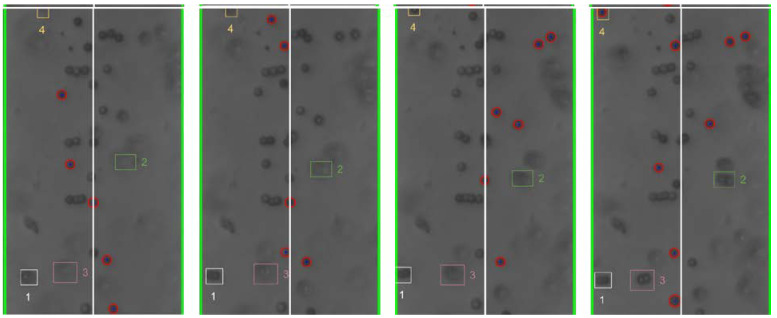
Individual bead movement during Trial 2 from frames 735 to 812.

**Figure 12 micromachines-13-00399-f012:**
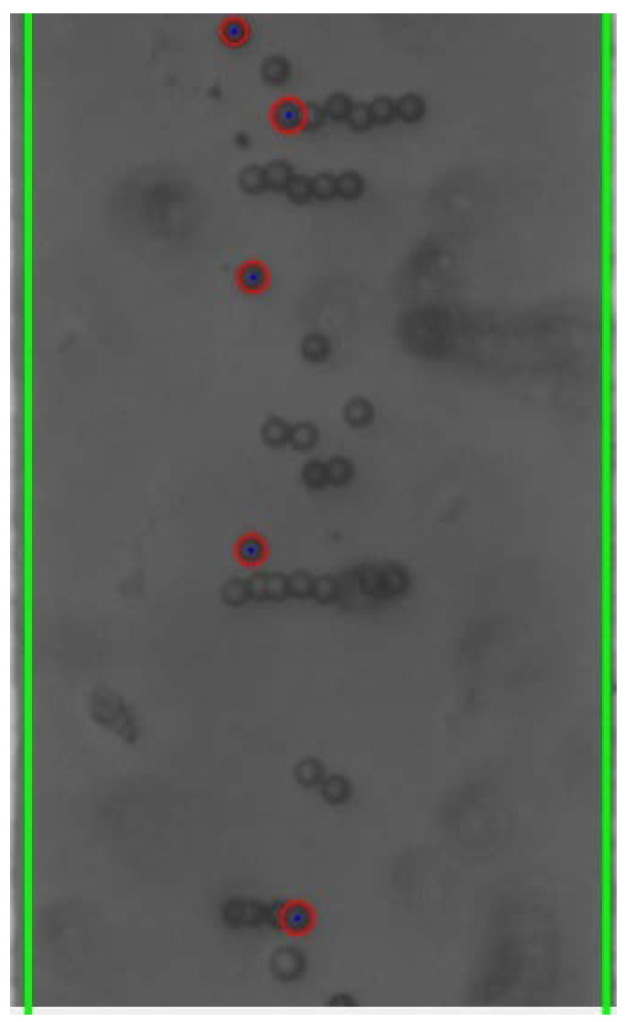
Pearl chain formation as a result of bead-to-bead interaction.

## Data Availability

The data presented in this study are available in Artificial Intelligence Algorithms Enable Automated Characterization of the Positive and Negative Dielectrophoretic Ranges of Applied Frequency and supporting Appendix A.

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
