# Peer review of "Artificial Intelligence Algorithms Enable Automated Characterization of the Positive and Negative Dielectrophoretic Ranges of Applied Frequency"

_micromachines, 2022, doi:10.3390/mi13030399_

Round 1

Reviewer 1 Report

In this draft paper, the authors proposed an AI algorithm assisted system that can be implemented with a microfluidic setting to investigate the DEP response against microbeads. The application is definitely helpful under certain scenarios, and embracing AI into the microfluidic-based investigation can always add the value to minimize the manual efforts. However, several conclusions are not well supported with evidence and some additional materials might be required to better elaborate the results. I think the manuscript could be suitable for publication in Micromachines after addressing my concerns bellow.

  1. Introduction part: the rationale about how the developed DEP analyzer can be applied on the micro- and nano- assembly remained obscure. Actually, in microfluidic setting DEP is quite useful for a lot of other aspects. For example, for analyzing the biological components from target cell & perform screening (ORG/10.1002/PLD3.11); separating cells from unwanted cells or reagents (DOI: 10.1039/d0lc00710b); fast CTC diagnosis (10.3390/CANCERS6010545), etc. Please consider revising this section or provide more reference on how DEP characteristics is helping the micro assembly process.
  2. CM factor calculation: the Clausius Mossotti factor is always dependent with background medium, and it can be totally different due to the dielectric properties of beads that you choose. In some case, beads can exhibit nDEP at all frequency. Hence, it will be much better to actually calculate the beads response using equation in section 2 and determine the actual crossover frequency as a benchmark. Example can be found in DOI: 10.1039/d0lc00710b.
  3. Algorithm design (section 3.3 & 3.4): in these two sections, section 3.3 uses the x axis displacement due to the nature of DEP is working on the perpendicular direction. However, moving into the section 3.4, the total judgement of whether the cell will be analyzed is with total planar displacement. Why would it be changed at this section?
  4. Particle detection (section 4.1), in this section, author mentioned “larger beads with lower concentration will result in better detection”. Any supporting material? And why decision made on using 3um beads among the tested groups?
  5. Data interpretation (Section 4.2): “sharp” decrease is not very obvious among both trails, especially when comparing with trail #2, from 500kHz to 190kHz. This conclusion requires more evidence. It could be either more repeats or slower sweeping. And, base on my own experience, it would be better to switch on/off of the applied signal to observe clearest trajectory of beads, since continuously sweeping the frequency might have some delays on the bead movement due to their momentum, thus potentially mislead the analysis of the DEP response.
  6. Section 4.4.2: crossover frequency conclusion is not well supported by comparing the previous studies, as we covered in point #2. Please consider revising this section accordingly by adding CM calculation materials.
  7. Beads not highlighted in Fig.8 with red circle. So, the AI provided cannot distinguish the same particle over different time frame – do you think this discrepancy might lead to a bias in analysis? Afterall, 20~30% is less than being the majority. Can you provide a dataset showing the rest of undetermined beads exhibited a similar trajectory profiles even they are not picked up by AI algorithm?

Minor:

  1. Section 2.1, exposure intensity is less important, please include the total dosage used.
  2. Figure 1 missing scale bar
  3. Supplementary material mentioned in section 4.2 is not included

Author Response

Please see the attached letter containing our responses to your review.

Thank you!

Reviewer 2 Report

The paper is very well organized and presented. The authors present and explain very well the physical problem as well as the algotithms they employ.

There are some minor corrections that should be made

1) Equations (1) and (2) are written in latex style and are not well viewed in the pdf file.

2) on page 6, 1st paragraph of 3.3 It should be with t as an index

Author Response

Please see the attachment for our response to your review.

Thank You!

Round 2

Reviewer 1 Report

Most suggestions have been resolved. Good to be published.